# Physical Performance, Anthropometrics and Functional Characteristics Influence the Intensity of Nonspecific Chronic Low Back Pain in Military Police Officers

**DOI:** 10.3390/ijerph17176434

**Published:** 2020-09-03

**Authors:** Janny M. A. Tavares, André L. F. Rodacki, Francielle Hoflinger, Alexandre dos Santos Cabral, Anderson C. Paulo, Cintia L. N. Rodacki

**Affiliations:** 1Department of Physical Education, Federal Technological University of Paraná, Universidade Tecnológica Federal do Paraná-UTFPR, Pedro Gusso Street, Neoville, 2601, Curitiba, Paraná CEP 81310-300, Brazil; janny@tavares.net.br (J.M.A.T.); andersoncaetano@gmail.com (A.C.P.); cintiarodacki@gmail.com (C.L.N.R.); 2Department of Physical Education, Centro Politécnico–, Universidade Federal do Paraná-UFPR, Rua Coronel Francisco H. dos Santos, 100, Jardim das Américas, Curitiba, Paraná CEP 81530-000, Brazil; 3Military Police Academy of Guatupê Curitiba/PR, Polícia Militar do Paraná-PMPR, BR-277 Km 72-Afonso Pena, Curitiba, Paraná CEP 81530-245, Brazil; franedfisica16@gmail.com; 4Department Medical, Curitiba/PR, Medical Department of Paraná Military Police Officers, Polícia Militar do Paraná-PMPR, Avenue Prefeito Omar Sabbag, 894, Jardim Botânico, Curitiba, Paraná CEP 80210-000, Brazil; cmacabral@casamilitar.pr.gov.br

**Keywords:** trunk strength, trunk resistance, chronic low back pain

## Abstract

*Background:* Chronic low back pain (CLBP) is a serious problem in Military Police Officers (MPO), which accounts for up to 45% of the sick leave rates. It has been assumed that the strength and the endurance of trunk flexor and extensor muscles are CLPB key factors, but it is not known whether these attributes are related to pain intensity. It was aimed to determine whether the strength and endurance of trunk flexor and extensor muscles differ in MPO with no pain (CON; *n* = 24), moderate (MOD; *n* = 42), and severe (SEV; *n* = 37) nonspecific chronic low back pain (CLBP). *Methods:* The peak torque and endurance test of trunk flexor (PTF.BM^−1^) and extensor (PTE.BM^−1^) muscles were compared. A multiple regression analysis was used to identify pain intensity predictors in all groups (PAIN) and according to pain intensity (MOD and SEV). *Results:* The PTF.BM^−1^ was negatively related to pain and was a significant predictor, irrespective of pain intensity (PAIN). *Conclusion:* When pain intensity was considered the PTF.BM^−1^ and PTE.BM^−1^ explained the pain in the MOD, while the PTE.BM^−1^ and service time explained pain intensity in the SEV. Endurance of the flexor and extensor muscles was not related to pain intensity. These results indicated that training protocols must emphasize specific strengthening routines.

## 1. Background

Chronic low back pain (CLBP) is a worldwide health problem that affects 85–90% of the adult population at some point in life [1]. The CLBP is characterized by persistent pain lasting three months or more, involving high treatment costs, lost productivity, and work absenteeism [2]. Military Police Officers (MPO) are subject to long working shifts, with high physical demands that are further increased by the overload of compulsory protective equipment (up to 14.0 kg). Indeed, MPO present high CLBP prevalence (from 41.0% to 43.6%) [3] and high sick leave rates (up to 45%) [4]. On the other hand, MPOs are considered as physically fit and subjected to intense physical training programs and testing.

Reduced stability and diminished postural control efficiency are known as a CLBP predisposing factor that relies on trunk flexor and extensor muscles to produce and sustain torque [5]. However, there are conflicting results regarding the ability to create a torque of the trunk flexor and extensor muscles and CLBP [6]. For example, the strength of the trunk flexor and extensor muscles is described as lower (25% and 35%, respectively) in CLBP individuals than in asymptomatic counterparts [6]. On the other hand, no differences between the torque of the extensor muscles of CLBP and healthy matched individuals were found [7]. Besides, the strength ratio between the trunk extensor and flexor muscles is lower than that reported in asymptomatic individuals [8]. Thus, torque changes differ between CLBP and healthy counterparts [9].

The ability to produce large peak torques is relevant, especially when the trunk muscles are fully recruited to respond to a single maximal movement (e.g., lifting a load). On the other hand, little is known regarding the ability to sustain a load or maintain a posture for prolonged periods. Maintaining a posture or providing spinal stability does not require maximal, but rather low to moderate activation of the paraspinal and abdominal muscles [10]). Exercises designed to treat CLBP require low activation levels ranging between 13% and 35% of the maximal isometric voluntary contraction [11]. Thus, the ability to sustain torque (endurance or resistance) is relevant. Therefore, it is unclear whether CLBP intensity is associated with the ability of the trunk muscles to generate maximal or to sustain torques (endurance).

Pain intensity is a limiting factor in producing maximum torques and sustaining loads. Henriksen and colleagues reported reduced knee extensors torques following an isotonic saline injection [12]. They also demonstrated increased generalized muscle inhibition in subjects with higher pain intensity, which may influence the ability to produce and sustain torque. It has been suggested that pain intensity is consistently and negatively associated with the physical performance during functional capacity assessments [13]. Therefore, individuals with higher pain intensity may present reduced physical performance, especially regarding the ability of the trunk flexor and extensor muscles to produce and sustain torque. Besides, individuals with a higher pain intensity may have worse physical performance as they are prone to exercise less and tend to avoid strenuous training. It has been demonstrated that patients with higher pain intensity deliberately attempt to suppress or prevent unwanted painful experiences [14]. Thus, it is likely that pain intensity may modulate the physical performance of the trunk flexor and extensor muscles.

This study aimed to investigate the relationship between the ability to produce (peak) and to sustain (endurance) torque of the trunk flexor and extensor muscles in MPO with ongoing moderate and severe pain intensity.

## 2. Methods

The study included male MPOs, active personnel of the MPO service of the Paraná State, who responded to an electronic call (Intranet) from the general headquarters. The number of individuals contacted was approximately 400 MPOs, from which 140 responded to the invitation. The inclusion criteria involved: (a) age between 21 and 55 years; (b) male; (c) free from injuries or diseases that might impede physical testing; (d) free from the pain that hampers the execution of the tests. Individuals with acute CLBP episodes, unable to perform physical tests were not included. Thus, after applying the inclusion/exclusion criteria, 103 participants were included in the study. The exclusion of 37 participants occurred due to the following reasons: schedule issues for testing (*n* = 32); previous surgery (*n* = 3); unable to perform the tests (*n* = 2). All participants were regularly attending the physical conditioning training program of the Military Academy, which included three sessions per week of 2 h. Participants signed an Informed Consent Form that had been approved by the Ethics Committee of the Paraná Federal Technological University, under number 2.214.386, which is in accordance with the Declaration of the World Medical Association.

### 2.1. Data Collection

In the first visit to the laboratory, participants answered a Visual Analogic Pain Scale (VAS) to quantify the pain intensity [15]. The pain scale is graded along a horizontally graduated line ranging from zero to ten, being classified as no or light pain (from 0 to 2.9), moderate pain (from 3.0 to 5.9), and severe pain (6.0 to 10). For analysis purposes, participants were separated into three groups according to pain intensity. Participants with no or light pain were allocated in the control group (CON group; *n* = 24). Participants with moderate pain were allocated to the moderate pain group (MOD group; *n* = 42); those with higher scores were assigned to the severe group (SEV group; *n* = 37). Service time was defined as the time of first registration, as reported in service records.

Participants were assessed for body mass and stature. The body mass index was calculated. The strength and endurance tests of trunk flexor and extensor muscles were performed in a random order. Participants received verbal encouragement and were asked to report adverse pain signs. No discomfort prevented participants from concluding the tests. They were confident that a maximal performance was achieved in all trials. A 10-min interval was used between tests and 3 min between each of the three trials. The best performance was used for analysis purposes.

The endurance test of the flexor muscles (ETF) was performed in a seated position [16], with the trunk flexed and supported at 60° backward, knees and hips flexed, and the feet secured between 0.30 and 0.45 m away from the buttocks. During the test, the participant was requested to sustain the position as long as possible. The trial ended when the initial position could not be sustained or by discomfort that prevented its continuation. The test is presented in Figure 1.

The endurance test of trunk extensors (ETE) was first proposed by Biering–Sorensen [17] and has predictive and discriminative validity for CLBP, with high reliability [18]. Participants were instructed to maintain their trunk above the base of support. The lower body was firmly secured with Velcro straps. Participants were requested to sustain the trunk horizontally aligned, as long as possible. The test was interrupted when the position could not be maintained or by discomfort. Figure 1—upper right panel shows the test’s position.

The peak torque of the trunk extensors (PTE) and flexor muscles (PTF) was defined as the highest torque produced in three maximal isometric attempts. The peak torque was determined using a load cell (EMG Systems, Brazil) with a resolution of 0.01 kgf, anchored to the trunk at pectoral height by an adjustable Velcro strap. The load cell (Figure 1—lower panels) was attached to a steel cable and secured on the ground. The torque was calculated as the product of the force by the distance of the fixation point to the center of the hip [19]. Postures are shown in Figure 1—lower right panel. The PTF and PTE were normalized for the body mass (PTF.BM^−1^ and PTE.BM^−1^, respectively). The ratio between the PTE/PTF was calculated. Pain intensity was also assessed immediately before the peak torque and endurance tests. The small variations between the instants in which pain was assessed were not large enough to demand reallocating a participant to another group. No participant reported a pain that impeded their maximal performance.

### 2.2. Statistical Analysis

Data normality was confirmed by the Kolmogorov–Smirnov test. A number of one-way ANOVAs for independent measurements were applied to compare age, service years, body mass, stature, BMI, peak torque, and endurance of the trunk flexor and extensor muscles, among CON, MOD, and SEV. The Bonferroni test was applied to identify where differences occurred. The effect size was also calculated. The association between pain intensity (EVA) and the predicting variables (age, service time, BMI, and the normalized peak torque and endurance) was determined using multiple regression analysis. The EVA used as a dependent variable was that measured immediately before the peak torque and endurance tests. The multiple regression analysis was first performed, including all participants with CLBP, irrespective of pain intensity (PAIN). Then two separate multiple regression analyses were performed to identify the pain intensity predicting variables (MOD and SEV). The CON was not included as they were not experiencing pain. The dependent variables presented a correlation bellow 0.7 to be included in the model, except age and service time, which are naturally correlated (r = 0.85). Therefore, multicollinearity was controlled. The Levene’s test was applied in the dependent variables and returned non-significant coefficients; thus, the homoscedasticity assumption was confirmed. The statistical procedures were performed using the Wizard Pro statistical package (version 1.9.38) and the significance level set at *p* < 0.05.

## 3. Results

No differences were found in age, stature, and BMI among groups. Body mass differed among groups, as the SEV was heavier than the other two groups. The CON’s service time was shorter than the other groups; no differences were found between MOD and SEV. The physical characteristics and performance are presented in Table 1.

The PTE and PTF were similar among groups. However, when the PTF was normalized by body mass, differences between the CON and SEV groups emerged.

The ETF was the greatest in the CON, intermediate in the MOD, and the lowest in the SEV group, while the ETE differed between the CON and SEV. When the endurance was normalized by the body mass, only the CON differed from the MOD and SEV groups. A large ES was found in all cases.

The regression analysis was performed for all participants with MOD and SEV pain, i.e., a group formed only by participants with CLBP, irrespective of the pain intensity (PAIN). The analysis showed a low ability to predict pain intensity in CLBP patients and explained approximately 20% of the pain intensity (r^2^ = 0.198). The PTF.BW^−1^ was the only significant predictor, which was negatively related to pain intensity. The analysis of the MOD group also showed a low ability of the predicting variables to explain pain intensity in participants with moderate pain (35%; r^2^ = 0.355). The PTE. BW^−1^ was a significant predictor. However, the PTE was positively associated with pain intensity. The ability of the predicting variables to explain pain intensity was around 50% (r^2^ = 0.50) in the SEV group. The service time was negatively associated with pain intensity. It was interesting to observe that the PTE.BW^−1^ was negatively related to pain intensity. The regression analysis results are presented in Table 2.

## 4. Discussion

This study aimed to determine the relationship between peak torque and endurance of the trunk flexor and extensor muscles in MPO with moderate (MOD) and severe (SEV) CLBP pain intensity. It was also aimed to observe if peak torque and endurance of the trunk flexor and extensor muscles were able to predict pain intensity in each group. The hypothesis that peak torque and endurance of the trunk flexor and extensor muscles normalized by body mass differ according to pain intensity groups was confirmed, except for the peak torque of the extensor muscles, which was similar among groups. The regression analysis showed a low ability to predict pain intensity in the group of moderate pain (35%), which was explained by 50% in the group of severe pain.

Although CLBP pain intensity was different between groups (2.5 points average difference), age and stature did not differ between groups. Even though age has been associated with CLBP [9], it is likely that the military police officers were still not under age-related degenerative effects as they were, on average, below 40 years old. Besides, age differences between groups (2–4 years) may not be clinically meaningful. Indeed, reports are indicating that CLBP prevalence is three to four times higher in older individuals when compared to younger ones [20]. Therefore, age was neither able to predict pain nor its intensity.

It is interesting to observe that the group with no CLBP (CON) presented the shortest service time in comparison to the two groups with CLPB (MOD and SEV). Service time was negatively associated as a pain intensity predictor only in the group with severe intensity, but not in the group with moderate pain intensity. Thus, pain intensity seems to be more critical in MPO exposed longer to the workloads, i.e., with longer service times.

Even though there is a linear correlation between CLBP and BMI in obese subjects [21], no differences were observed in the present study, as most participants presented an overweight profile but were not severely obese. Thus, the body overweight and the requirements of wearing heavy apparel may promote overloading of the articular structures of the spine, which become predisposed to degeneration and increased pain intensity. The chronic overloading caused by the excessive masses of the body has been indicated as a predisposing factor [22]. Indeed, body mass was larger in the group with severe pain. On the other hand, BMI was not able to predict pain intensity. The aggregated effect caused by large body masses and the weight of the safety apparel may explain better pain intensity than as individual predictors.

BMI was not a predictor of pain or pain intensity. It may be explained as obesity is more likely to occur in sedentary individuals, which is not the case of the present study as MPOs are frequently submitted to training programs. Obese and sedentary are expected to present muscle weakness, which is a risk factor for chronic and acute back pain [22]. Besides, BMI does not consider muscle mass and fat tissue differences, which are better reflected by the strength assessments (peak torque and resistance) of the trunk flexor and extensor muscles. Indeed, peak torque and endurance of the flexor and extensor muscles differed among groups, especially for the group with no pain. The group with no pain showed the best muscle performance when compared to the groups with moderate and severe pain.

The ability to generate large torques by the flexor muscles in healthy individuals in comparison to those with CLBP is a common finding. It was evident in the group with no pain, which showed 31% more ability to generate torque in the flexor muscles than the group with severe pain. The higher capacity of the flexor muscles has been associated with augmented intra-abdominal pressure, which is one of the mechanisms related to improved spinal stability [23]. The negative relationship between the peak torque of the flexor muscles and pain intensity supports the relevance of training programs designed to improve the strength of the abdominal muscles, even in supposedly well-conditioned individuals, such as MPOs. On the other hand, pain intensity was not predicted by the torque of the flexor muscles in the group with severe pain intensity. Indeed, improving the strength in individuals with ongoing CLBP must be a difficult task as it requires extensive muscle recruitment of specific sensitive and painful sites.

The peak torque of the extensor muscles was positively correlated with the pain in the MOD group. In other words, the higher the ability to produce a torque of the extensor muscles, the higher is the CLBP pain intensity. It has been argued that excessive training routines in firefighters may have aggravated the potential for injuries, as they may have an increased exposure to musculoskeletal problems [24]. In addition, it was shown that although the more well-conditioned firefighters had fewer injuries, their injuries were more severe. Brown and colleagues [25] argued that fitter individuals may also complain less about smaller injuries. McGill and colleagues [10] have argued in favor of an optimal level of fitness for injury resilience, in which a low level of physical conditioning may cause injuries as they have not been trained to meet the demand. Paradoxically, a high level of physical conditioning, in conjunction with large demands from the continued job requirements, may result in cumulative trauma to run ahead of the rate of repair. The overweighed status of the participants may have contributed to an increasing chronic exposure to heavy workloads^3^. These arguments may explain the positive association between the peak torque of the trunk extensor muscles and pain intensity in the moderate pain intensity group. It is also plausible that the stronger extensor muscles may increase the compressive forces applied to several structures of the spine that are not designed to bear large forces [22].

On the other hand, the negative association between the peak torque of the extensor muscles and pain intensity in the group with severe CLBP is intriguing. It may be speculated that pain intensity may cause different responses in CLBP patients. It has been demonstrated that subjects with pain are able to develop alternate postural adjustments as a strategy performed to cope with the pain. These adjustments are accompanied by muscle recruitment changes in which some muscle groups are selectively recruited according to a specific pattern. Lorimer and Hodges demonstrated a delayed recruitment of the transversus abdominis/obliquus internus coupled with an increased recruitment of the obliquus externus during a set of trials in which a painful cutaneous stimulation was applied [26]). A gradual return to control values when the pain ceased was noticed. Thus, other mechanisms may be involved that allow one to increase the pain perception threshold by activating pain modulation mechanisms [27]. Chronic exposure to severe pain may have produced central adaptations, which require future investigations that are beyond the scope of the present study.

The ability of the trunk muscles to sustain force for prolonged periods is essential to providing spinal stability. It must be noted that spinal stability does not emerge from the ability of the muscles to produce maximal torques but on its ability to sustain low and moderate torques over time. Indeed, rehabilitation protocols have included exercises that do not exceed 20% to 30% of the maximal activation [28], i.e., they provide minimal strength training effect. The group with no CLBP sustained the load applied to trunk longer than the groups with CLBP. On the other hand, the ability to resist fatigue (i.e., sustain a low to moderate torque) was able to predict CLBP, but not its intensity. Poor endurance performance of the extensor muscles of the trunk is a marker for future low back problems [17,29]. The ability of the trunk extensors to sustain torque was lower than that reported in the literature [30]. McGill and colleagues [10] reported 131.3 s for individuals without CLBP, which was almost twice the time the MPOs with severe pain sustained the position. It indicates that the MPOs’ sustained force of the extensor muscles was very low in comparison to normative references.

Although the endurance of the flexor and extensor muscles was relevant when comparing individuals with and without CLBP, the extensors showed a smaller ability to sustain the load of the trunk segment in comparison to the flexor muscles. A low trunk extensor activity has been reported while performing stabilizing exercises compared to the abdominals [31], which may explain the differences among muscle groups. On the other hand, the differences between the groups with moderate and severe pain intensity and the group with no pain were larger for the flexor (from 38% to 49%) than the extensor muscles (19% to 37%). It may suggest that the endurance of the flexors is more relevant than the extensor muscles, especially in occupations that are extenuating, and incorporate additional body mass loading (e.g., additional protective apparel), such as MPOs.

The regression equations presented a low capacity to predict pain intensity (i.e., 20%), and a substantial amount of variation was not explained when peak torque and endurance of the trunk flexor and extensor muscles were associated with a few anthropometric and functional aspects of MPOs. The multifactorial and complex nature of the CLBP may require an inclusion of other domains such as muscle activation and muscle coordination, which may also play a significant role. Besides, individual perception of pain may have also played a role. Thus, estimating pain intensity may require a larger number of domains that are not included in the present study (e.g., perceptual aspects, muscle coordination, etc.).

The reader should bear in mind some limitations of the study that must be viewed with caution while interpreting the results. It is not possible to ensure that all participants performed the peak torque or endurance tests maximally, although there were no reports of disabling pain that impeded the execution of the tests. Besides, the time participants were sustaining CLBP was not controlled and may have influenced the way they cope with discomfort and pain. It is difficult to monitor whether participants with CLBP avoided some particular (painful) activities and if they were reallocated to other less demanding duties in any period of their service history. The influence of age is also a limiting factor, although differences among groups were relatively small (3–4 years) and may be viewed as little from a clinical point of view. Another limiting factor is that some parameters such as BMI, which may be a CLBP aggravating factor, is caused by large body masses that are chronically applied to the spine [22]. Finally, the study did not include female participants.

## 5. Conclusions

This study aimed to verify whether the strength and endurance of the flexor and extensor muscles of the trunk are related to CLBP intensity in MPOs. The major finding was that peak torque of the trunk extensor muscles differed between participants with severe pain intensity and no pain. The peak of the extensor muscles did not vary between individuals without pain in comparison to those with moderate or severe pain, irrespective of the pain intensity. The endurance of the extensor and flexor muscles was higher in individuals without low back pain, but it did not explain pain intensity.

## Figures and Tables

**Figure 1 ijerph-17-06434-f001:**
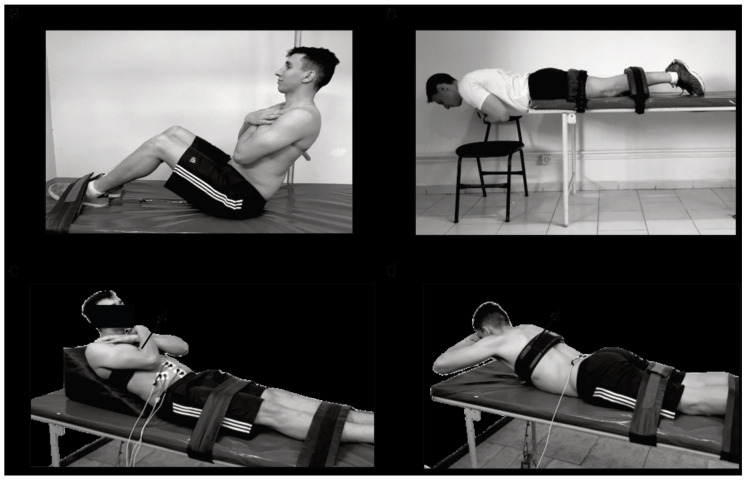
Representation of the assessment of the endurance test of the trunk flexor (upper left panel), and extensor (upper right panel) muscles, and the peak torque of the trunk flexor (lower left panel) and extensor (lower right panel) muscles.

**Table 1 ijerph-17-06434-t001:** Physical characteristics and performance on peak torque and endurance tests of the trunk flexor and extensor muscles of the control (CON) group, and the groups with moderate (MOD) and with severe (SEV) chronic low back pain (CLBP).

Variables	CON (*n* = 24)	MOD (*n* = 42)	SEV (*n* = 37)	*p*	ES
Age (years)	34.7 ± 5.4	38.3 ± 6.6	38.0 ± 7.4	0.083	0.54
Service Time (years)	9.6 ± 4.7 ^2,3^	15.1 ± 8.4 ^1^	14.0 ± 8.21	**0.018**	0.70
VAS (a.u)	0.6 ± 0.9 ^2,3^	4.2 ± 0.9 ^1,3^	6.5 ± 0.7 ^1,2^	**<0.010**	2.49
Stature (m)	1.73 ± 0.05	1.72 ±0.06	1.75 ± 0.04	0.143	0.41
BM (Kg)	80.2 ± 11.0 ^3^	81.6 ± 10.9 ^3^	86.9 ± 10.4 ^1,2^	**0.032**	0.60
BMI (kg.m^−2^)	26.6 ± 3.1	27.2 ± 2.5	28.2 ± 3.1	**0.079**	0.56
PTF (N.m)	240.0 ± 61.0	236.2 ± 47.0	223.0 ± 75.6	0.531	0.27
PTE (N.m)	157.2 ± 60.9	187.6 ± 46.6	186.8 ± 63.8	0.080	0.52
ETF (s)	138.4 ± 62.4 ^2,3^	87.6 ± 61.4 ^1,3^	76.2 ± 46.1^1,2^	**<0.010**	1.08
ETE (s)	100.4 ± 35.2 ^3^	82.2 ± 39.6	67.6± 28.9 ^1^	**<0.010**	0.89
PTF.BM^−1^ (N.m.Kg^−1^)	3.05 ± 0.92 ^3^	2.91 ± 0.51	2.55 ± 0.74 ^1^	**0.017**	0.68
PTE.BM^−1^ (N.m.Kg^−1^)	2.00 ± 0.89	2.32 ± 0.71	2.13 ± 0.66	0.192	0.45
ETF.BM^−1^ (s.Kg^−1^)	1.76 ± 0.88 ^2,3^	1.09 ± 0.78 ^1^	0.89 ± 0.53 ^1^	**<0.010**	1.10
ETE.BM^−1^ (s.Kg^−1^)	1.27 ± 0.45 ^2,3^	1.02 ± 0.51^1^	0.79 ± 0.35 ^1^	**<0.010**	1.02

VAS = Visual Analogic Scale; BM = Body Mass; BMI = Body Mass Index; PTF = Peak torque of the trunk flexors; PTE = Peak torque of the trunk extensors; ETF = Endurance of the trunk flexors; ETE = Endurance of the trunk extensors; PTF.BM^−1^ = PTF Normalized by BM; PTE.BM^−1^ = PTE Normalized by BM; ETF.BM^−1^ = ETF Normalized by BM; ETE.BM^−1^ = ETE Normalized by BM. *p* values highlighted in bold are significant. ^1^—refers differences with respect to CON; ^2^—refers differences with respect to MOD; ^3^—refers differences with respect to SEV.

**Table 2 ijerph-17-06434-t002:** Regression analysis with the predicting variables with moderate (MOD) and severe (SEV) pain and both groups, irrespective of pain intensity (PAIN).

	Variables	Std Coef.	Std. Error	95% Conf. Interval		*t*	*p*
PAIN (*n* = 79)	Age (years)	−0.008	0.048	−0.104	0.089	−0.156	0.876
	BMI (kg.m^−2^)	0.033	0.062	−0.090	0.156	0.534	0.595
	Service Time (years)	−0.021	0.040	−0.101	0.059	−0.531	0.597
	PTF.BW^−1^ (N.m.BW^−1^)	−0.847	0.312	−1.470	−0.224	−2.712	**0.008**
	PTE.BW^−1^ (N.m.BW^−1^)	0.183	0.301	−0.417	0.783	0.609	0.545
	ETF.BW^−1^ (s.BW^−1^)	0.028	0.352	−0.674	0.730	0.080	0.936
	ETE.BW^−1^ (s.BW^−1^)	−0.828	0.548	−1.922	0.265	−1.511	0.135
	*Constant*	7.646	2.740	2.183	13.109	2.791	0.007
MOD (*n* = 42)	Age (years)	−0.006	0.044	−0.095	0.083	−0.136	0.892
	BMI (kg.m^−2^)	0.049	0.059	−0.07	0.168	0.834	0.410
	Service Time (years)	0.011	0.034	−0.058	0.081	0.330	0.744
	PTF.BW^−1^ (N.m.BW^−1^)	−0.624	0.297	−1.228	−0.02	−2.099	**0.043**
	PTE.BW^−1^ (N.m.BW^−1^)	0.836	0.255	0.318	1.354	3.282	**0.002**
	ETF.BW^−1^ (s.BW^−1^)	−0.100	0.270	−0.648	0.448	−0.371	0.713
	ETE.BW^−1^ (s.BW^−1^)	−0.317	0.419	−1.168	0.535	−0.755	0.455
	*Constant*	3.219	2.650	−2.167	8.604	1.214	0.233
SEV (*n* = 37)	Age (years)	0.038	0.029	−0.022	0.097	1.293	0.206
	BMI (kg.m^−2^)	0.001	0.035	−0.072	0.071	−0.010	0.992
	Service Time (years)	−0.051	0.025	−0.103	0.001	−2.049	**0.050**
	PTF.BW^−1^ (N.m.BW^−1^)	−0.025	0.180	−0.344	0.394	0.137	0.892
	PTE.BW^−1^ (N.m.BW^−1^)	−0.630	0.182	−1.003	−0.257	−3.452	**0.002**
	ETF.BW^−1^ (s.BW^−1^)	0.160	0.255	−0.363	0.682	0.625	0.537
	ETE.BW^−1^ (s.BW^−1^)	0.162	0.402	−0.660	0.983	0.403	0.690
	*Constant*	6.904	1.581	3.671	10.138	4.367	0.000

VAS = Visual Analogic Scale; BM = Body Mass; BMI = Body Mass Index; PTF = Peak torque of the trunk flexors; PTE = Peak torque of the trunk extensors; ETF = Endurance of the trunk flexors; ETE = Endurance of the trunk extensors; PTF.BM^−1^ = PTF Normalized by BM; PTE.BM^−1^ = PTE Normalized by BM; ETF.BM^−1^; = ETF Normalized by BM; ETE.BM^−1^ = ETE Normalized by BM. *p* values highlighted in bold are significant.

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
