# Peer review of "Physical Performance, Anthropometrics and Functional Characteristics Influence the Intensity of Nonspecific Chronic Low Back Pain in Military Police Officers"

_ijerph, 2020, doi:10.3390/ijerph17176434_

Round 1
Reviewer 1 Report
Overall, the methodology of the assessment of chronic low back pain, peak torque and endurance of groups of muscles does not raise reservation. However, the approach to the analysis is hardly understandable, and authors do not provide a rationale why they performed the analysis in the subgroups distinguished based on the independent variable further used in specific linear regression models. Apart from the fact, that such strategy is not explained rationally, in the result of such manipulation the number of cases in the specific models becomes very low and it is hardly possible to proceed with valid conclusions about the importance of particular predictors.
Actually, it is also not clear why they have excluded participants who report no or low-level symptoms of back pain in the VAS.
Multivariate linear regression models performed in the MOD and SEV subsamples counting about 30-40 subjects and with seven independent variables may result in an artificial overestimation of the effects of singular measures.
Furthermore, it may be seen (table 1) that younger persons were less prone to report no or low CLB pain with VAS. But, atomising the analysis results in a paradoxical finding that age is not associated with the level of pain.
To sum up, I believe that Authors should present the results of the multivariate multiply linear regression for all subjects assessing their level of pain with VAS. If there is a need to make the analysis after categorisation of the level of CLBP, they should perform multivariate multiply logistic regression after dichotomising CLBP based on the VAS scores.
Apart from the doubts about the general approach to the analysis, one can see that no assumptions for the model of linear regression like homoscedasticity, autocorrelation, are reported in the paper.
Author Response
REVIEWER #1
Overall, the methodology of the assessment of chronic low back pain, peak torque and endurance of groups of muscles does not raise reservation. However, the approach to the analysis is hardly understandable, and authors do not provide a rationale why they performed the analysis in the subgroups distinguished based on the independent variable further used in specific linear regression models. Apart from the fact, that such strategy is not explained rationally, in the result of such manipulation the number of cases in the specific models becomes very low and it is hardly possible to proceed with valid conclusions about the importance of particular predictors.
Actually, it is also not clear why they have excluded participants who report no or low-level symptoms of back pain in the VAS.
Multivariate linear regression models performed in the MOD and SEV subsamples counting about 30-40 subjects and with seven independent variables may result in an artificial overestimation of the effects of singular measures.
Furthermore, it may be seen (table 1) that younger persons were less prone to report no or low CLB pain with VAS. But, atomising the analysis results in a paradoxical finding that age is not associated with the level of pain.
To sum up, I believe that Authors should present the results of the multivariate multiply linear regression for all subjects assessing their level of pain with VAS. If there is a need to make the analysis after categorisation of the level of CLBP, they should perform multivariate multiply logistic regression after dichotomising CLBP based on the VAS scores.
Apart from the doubts about the general approach to the analysis, one can see that no assumptions for the model of linear regression like homoscedasticity, autocorrelation, are reported in the paper.
ANSWERS:
Initially, we want to use this opportunity to thank the reviewer for his time and efforts to provide us relevant feedback. We hope we have resolved all issues in our point-by-point response. The main issues raised by the reviewer were addressed bellow.
The reviewer is correct, and we recognize that a clear rationale was missing. We have included a couple sentences to explain why higher pain intensity may impact on physical performance of the trunk flexor and extensor muscles. The text read as follows:
“It has been suggested that pain intensity is consistently and negatively associated with physical performance during functional capacity assessments (Gross, 2006). Therefore, individuals with higher pain intensity may present reduced physical performance, especially regarding the ability of the trunk flexor and extensor muscles to produce and sustain torque. Besides, individuals with higher pain intensity may have worse physical performance as they are prone to exercise less and tend to avoid strenuous training. It has been demonstrated that patients with higher pain intensity deliberate attempt to suppress or prevent unwanted painful experiences (Kroska, 2016). Thus, it is likely that pain intensity may modulate physical performance of the trunk flexor and extensor muscles.”
The reviewer also questions the number of cases in each analysis would be critical. Therefore, we performed two separate analysis. The first one was performed considering pain intensity using two distinct groups (MOD and SEV). We are aware that this approach is limited as the number of participants is not expressive, although we have observed several studies in which similar approaches (i.e., regression analysis) are performed in considerably smaller samples. The second analysis incorporates all participants (referred as PAIN), in which the number of subjects is more palatable. We emphasize that this is a comprehensive approach considering the small number of independent variables (7 variables).
We have excluded the participants who reported low-level or no symptoms because they would bias the analysis as they would score 0 in the VAS. In addition, they main purpose of the study was to determine how pain intensity modulates physical performances. Thus, the inclusion of participants with no pain symptoms would be an additional confounding factor. We hope the reviewer understand that we have no intentions to include asymptomatic participants as they would bias the analysis. In fact, we have performed the analysis and noticed that none of the independent variables were significant, even though PTF.BW-1 was borderline (0.058).
|
EVA PARM |
EVA STD ERR |
EVA T |
EVA p |
-95% CI |
+ 95% CI |
EVA B |
EVA STD ERR |
-95% CI |
+95% CI |
Intercept |
4.543937 |
3.408068 |
1.33329 |
0.187933 |
-2.28598 |
11.37386 |
|
|
|
|
Age |
0.047884 |
0.055637 |
0.86064 |
0.393171 |
-0.06362 |
0.15938 |
0.235158 |
0.273235 |
-0.312418 |
0.782734 |
BMI |
0.063717 |
0.079353 |
0.80295 |
0.425458 |
-0.09531 |
0.22274 |
0.139033 |
0.173152 |
-0.207971 |
0.486037 |
Serv Time |
-0.053736 |
0.047306 |
-1.13593 |
0.260910 |
-0.14854 |
0.04107 |
-0.322094 |
0.283550 |
-0.890340 |
0.246153 |
PFT.BW-1 |
-0.632211 |
0.327254 |
-1.93186 |
0.058534 |
-1.28804 |
0.02362 |
-0.340373 |
0.176189 |
-0.693464 |
0.012717 |
PTE.BW-1 |
-0.192166 |
0.359396 |
-0.53469 |
0.595018 |
-0.91241 |
0.52808 |
-0.093376 |
0.174635 |
-0.443351 |
0.256600 |
ETF.BW-1 |
0.252281 |
0.521225 |
0.48401 |
0.630297 |
-0.79228 |
1.29684 |
0.074788 |
0.154516 |
-0.234868 |
0.384444 |
EFE.BW-1 |
0.095242 |
0.681288 |
0.13980 |
0.889331 |
-1.27009 |
1.46057 |
0.023577 |
0.168649 |
-0.314404 |
0.361558 |
Yes, we agree with the reviewer that age is a confounding factor, but we have no means of removing such effect. Age itself wasn’t a significant explanatory variable in our model (p values > 0.9), even though a significant difference in age was identified between groups. Note that such differences are not large and involve 3-4 years, which is not a meaningful difference. To accommodate the reviewer’s concern, we acknowledged this in the text, as follows:
“Besides, age differences between groups (2-4 years) may not be clinically meaningful”.
The reviewer is correct, and we have included additional information to provide a more comprehensive view of the regression analysis assumptions that reads:
“The dependent variables presented a correlation bellow 0.7 to be included in the model, except age and service time, which are naturally correlated (r=0.85). Therefore, multicollinearity was controlled. The Levene’s test was applied in the dependent variables and returned non-significant coefficients; thus, the homocedasticity assumption was confirmed.”
Reviewer 2 Report
Dear Authors,
I find the article interesting, however I would like to point some issues.
Study included people with age range 21-55 years. In context of chronic low back pain I find it inappropriate. The etiology of chronic low back pain differs due to age. In my opinion it might interfere groups homogeneity. As the participants were divided into three groups depending on pain intensity the age of the participants did not differ significantly however it is still unclear what were the reasons for chronic low back pain.
I recommend also to provide more detailed exclusion criteria.
Information about consent agreement is given twice. I suggest to correct that.
While using one-way ANOVA homogeneity of variance should be check with Leven's test. Authors should provide information about that. I find no need to use Kolmogorov-Smirnov test in that study.
BMI was not a predictor of pain intensity. Firstly, while writing about training programs authors ought to provide data about frequency of those training or at least assess the level of physical activity with IPAQ. Moreover, people taking part in the study were overweight due to mean value of BMI. As authors claim that BMI do not include assessment of fat and muscle mass it doesn't explain that in the study took part people with highly developed musculature. Taking into consideration fact of diminished basic metabolic rate in older people those participants might be simply overweight. Thus study protocol ought to include assessment of body composition or at least measurement of WHR. I recommend to point that in study limitations.
Author Response
REVIEWER#2
I find the article interesting, however I would like to point some issues.
Study included people with age range 21-55 years. In context of chronic low back pain I find it inappropriate. The etiology of chronic low back pain differs due to age. In my opinion it might interfere groups homogeneity. As the participants were divided into three groups depending on pain intensity the age of the participants did not differ significantly however it is still unclear what were the reasons for chronic low back pain.
I recommend also to provide more detailed exclusion criteria.
Information about consent agreement is given twice. I suggest to correct that.
While using one-way ANOVA homogeneity of variance should be check with Leven's test. Authors should provide information about that. I find no need to use Kolmogorov-Smirnov test in that study.
BMI was not a predictor of pain intensity. Firstly, while writing about training programs authors ought to provide data about frequency of those training or at least assess the level of physical activity with IPAQ. Moreover, people taking part in the study were overweight due to mean value of BMI. As authors claim that BMI do not include assessment of fat and muscle mass it doesn't explain that in the study took part people with highly developed musculature. Taking into consideration fact of diminished basic metabolic rate in older people those participants might be simply overweight. Thus study protocol ought to include assessment of body composition or at least measurement of WHR. I recommend to point that in study limitations.
ANSWERS:
Initially, we want to use this opportunity to thank the reviewer for his time and efforts to provide us relevant feedback. We hope we have resolved all issues in our point-by-point response. The main issues raised by the reviewer were addressed bellow.
We partially agree with the reviewer that age factor is inappropriate factor in chronic low back pain. Age itself wasn’t a significant explanatory variable in our model (p values > 0.9), even though age differed between groups. Note that such differences are not large and involve 3-4 years, which is not a meaningful difference. Besides, to isolate the effects of age we would have to discard an expressive number of participants. To accommodate the reviewer’s concern, we acknowledged this in the text, as follows:
“Besides, age differences between groups (2-4 years) may not be clinically meaningful”.
Additionally, we have also acknowledged that age as a limiting factor in the present study. To accommodate the concern of the reviewer, we have included a statement in the limitation section that reads:
“The influence of age is also a limiting factor, although differences between groups were relatively small (3-4 years) and may be viewed as smal from a clinical point of view. “
The reviewer also argues that it is still unclear what were the reasons for back pain. We tried to answer this question in the discussion, however, age was not a significant explanatory factor in any of the models we run. Thus, other factors were discussed to explain low back pain.
The reviewer is correct, and we have removed the duplicate information regarding the informed consent procedures and the very end of the methods section.
We partially agree with the reviewer. We disagree that the IPAQ is a proper instrument for such assessments. It has been demonstrated that such questionnaire does not correlate with some specific physical performance measures. Please, note that comparisons between podometers and the scores obtained using the IPAQ hold are described to vary from low to moderate (Benedeti et al., Reproducibility and validity of the International Physical Activity Questionnaire (IPAQ) in elderly men. Rev Bras Med Esporte; Vol. 13, No 1 – Jan/Fev, 2007). Furthermore, the results from the IPAQ questionnaire are prone to produce a homogeneous classification as all participants were regularly engaged in physical conditioning activities (i.e., military training routines). Finally, questionnaires are predictive and do not represent specific measures of the actual conditioning of the specific movers and stabilizing muscles of the spine. A general measure of the physical activity level may not provide a more specific and detailed idea of the local functional status of the relevant muscles acting around the trunk.
The Levene’s test was performed and is included in the statistics section.
In addition, the reviewer questions the arguments used to explain BMI. We understand the point of the reviewer, however, BMI may represent a chronic loading condition but does not reveal the local and specific contribution of the trunk surrounding muscles. Therefore, we reorganized the discussion paragraph and also highlighted the limitations of BMI usage.
The insertions read as follows:
Included
“All participants were regularly attending the physical conditioning training program of the Military Academy, which included three sessions per week of 2 hours.”
Adjusted
“BMI was not a predictor of pain or pain intensity. It may be explained as obesity is more likely to occur in sedentary individuals, which is not the case of the present study as MPOs are frequently submitted to training programs. Obese and sedentary are likely to present muscle weakness, which is a risk factor for chronic and acute back pain (Rodacki et al. 2005). Besides, BMI does not consider muscle mass and fat tissue differences, which are better reflected by the strength assessments (peak torque and resistance) of the trunk flexor and extensor muscles. Indeed, peak torque and endurance of the flexor and extensor muscles differed between groups, especially for the group with no pain. The group with no pain showed the best muscle performance when compared to the groups with moderate and severe pain”.
Included
“Another limiting factor is that some parameters such as BMI, which may be a CLBP aggravating factor cause by large body masses that are chronically applied to the spine (Rodacki et al., 2005).”

Reviewer 3 Report
Thank you for the opportunity to evaluate this interesting manuscript. Overall, the idea of the study is very interesting. This study tackles important information for this field of study. I think that the present form is acceptable but the below should be added for details.
Title
- Please revise the title that can explain the text specifically.
Discussion
- Discussion has to start the hypothesis of the study and the results of data for your study.
- What were the strengths of this study?
Author Response
REVIEWER#3
Thank you for the opportunity to evaluate this interesting manuscript. Overall, the idea of the study is very interesting. This study tackles important information for this field of study. I think that the present form is acceptable but the below should be added for details.
Title
- Please revise the title that can explain the text specifically.
Discussion
- Discussion has to start the hypothesis of the study and the results of data for your study.
- What were the strengths of this study?
ANSWERS
Initially, we want to use this opportunity to thank the reviewer for his time and efforts to provide us relevant feedback. We hope we have resolved all issues in our point-by-point response. The main issues raised by the reviewer were addressed bellow.
The title was revised and changed accordingly. Indeed, we measured more than physical characteristics. It reads as follows:
“Physical Performance, Anthropometrics and Functional Characteristics Influence the Intensity of Nonspecific Chronic Low Back Pain in Military Police Officers”.
The start of the discussion section was modified to accommodate the reviewer’s concern and reads:
“The hypothesis that peak torque and endurance of the trunk flexor and extensor muscles normalized by body mass differ according to pain intensity groups was confirmed, except for the peak torque of the extensor muscles, which was similar among groups. The regression analysis showed low ability to predict pain intensity in the group of moderate pain (35%), which was explained by 50% in the group of severe pain.”
The strengths of the study include a large sample of MPOs that were assessed for their more relevant muscles operating around the spinal column. This is one of the first attempts to relate CLBP intensity with muscle functioning. It is also relevant to mention muscle functioning included the measures peak torque and endurance, which are not always present in most studies.

Reviewer 4 Report
This articles title “Physical Variables Explain the Intensity of Nonspecific Chronic Low Back Pain in Military Police Officers”, that contributed by Tavares and colleagues, proved the peak of the extensor muscle has not effect the pain intensity on the low back, and the endurance of extensor and flexor muscles show greater no pain group compare with other groups. A large amount of data is used in the article, and the number of test subjects is large. The research on special police has very good practical significance, the data is very clear and reliable,but at the same time, because the experimental subjects are military police officers, the research on low back pain cannot be compared with sex difference.
This manuscript is finished by Tavates and colleagues. Through a lot of experimental data to study the correlation between low back pain and muscles in MPO officers,
The highlights:
1 The experimental subjects were Military Police Office, and the impact of low back pain on their work and life was worthy of attention.
2 The sample quantity is large enough to provide effective data.
There are also some questions and Suggestions as follows:
1 Although the author's research object is the Military Police Office, is there many female objects be counting. The low back pain has a significant sex different, So the author should discuss it in discussion part.
2 The author ‘s result shows the service time has relationship with the low back pain, do you have any control groups like same age and other jobs (service time).
3 It is important that the authors take weight into account. My question is whether the author has considered the obesity problem, and the body fat rate at the same age may better reflect the real situation.
4 To avoid misunderstanding the result, the author should improve some English sentences.
Author Response
REVIEWER #4
This articles title “Physical Variables Explain the Intensity of Nonspecific Chronic Low Back Pain in Military Police Officers”, that contributed by Tavares and colleagues, proved the peak of the extensor muscle has not effect the pain intensity on the low back, and the endurance of extensor and flexor muscles show greater no pain group compare with other groups. A large amount of data is used in the article, and the number of test subjects is large. The research on special police has very good practical significance, the data is very clear and reliable but at the same time, because the experimental subjects are military police officers, the research on low back pain cannot be compared with sex difference.
This manuscript is finished by Tavates and colleagues. Through a lot of experimental data to study the correlation between low back pain and muscles in MPO officers,
The highlights:
1 The experimental subjects were Military Police Office, and the impact of low back pain on their work and life was worthy of attention.
2 The sample quantity is large enough to provide effective data.
There are also some questions and Suggestions as follows:
1 Although the author's research object is the Military Police Office, is there many female objects be counting. The low back pain has a significant sex different, So the author should discuss it in discussion part.
2 The author ‘s result shows the service time has relationship with the low back pain, do you have any control groups like same age and other jobs (service time).
3 It is important that the authors take weight into account. My question is whether the author has considered the obesity problem, and the body fat rate at the same age may better reflect the real situation.
4 To avoid misunderstanding the result, the author should improve some English sentences.
ANSWERS
Initially, we want to use this opportunity to thank the reviewer for his time and efforts to provide us relevant feedback. We hope we have resolved all issues in our point-by-point response. The main issues raised by the reviewer were addressed bellow.
Indeed, our sample did not include female participants. The selection of male participants was not intentional. To accommodate the reviewer’s concern, we have included such limitation at the end of the discussion section as it reads:
“Finally, the study did not include female participants.”
Unfortunately, we do not have controls with the same age or time of service.
We have taken body mass into account as we have normalized all peak torque and resistance measures normalized by body mass (i.e., body weight). In fact, obesity is a complex problem as obese participants experience a chronic loading and may be more affected than other participants with “normal” body weight (BMI). Although obesity may play a role, we acknowledged this factor, but preferred to be more focused on muscle functioning (peak torque and endurance) than in weight effects. We want to stress the argument that body mass effect was recognized as an additional effect. Age may be also an additional issue, however, and despite the fact that age differed between groups, such differences may not be of clinical significance, as only 3-4 years of differences were identified between groups.
We have revised the manuscript and are confident that all misspellings are resolved.

Reviewer 5 Report
It is a paper about authors’ research efforts to make clear the relationship between the chronic low back pain and peak torque, endurance of the trunk flexor and extensor muscles in military police officers (MPO). Experiments were done for collecting data from the MPO groups without pain, with moderate pain and with severe pain. Statistical test was performed to check the difference between the groups. And multivariate regression analysis was performed to identify the relationship between the pain and the target torque and endurance test values.
The data collection and the statistics used are appropriate. The results have been well presented. The topic should be of interest to the readers of the journal.
However, additional information of data collection process and data collected, and/or further analysis are required for interpretation of the results, which are important for the understating of the CLBP of MPO groups. Here are two concerns.
1) The strength and endurance tests might be seriously affected by the pain
This was noted by the authors themselves at the end of the Discussion section, too. Since the subjects were asked to report “adverse pain signs”, it should be possible for the authors to show the reported “adverse pain signs” for different pain groups. Especially for the severe pain group, the report during the tests, shall be very informative for the readers to understand the results.
And since the VAS of the pain and tests have been performed on different days, the pain perception on the day of the tests might be more important. Any information collected on the day of tests might help understand the real nature of the relationship between the pain and peak torque, endurance of the trunk flexor and extensor muscle.
2) Difference between moderate pain and severe pain groups needs further explanation
In the regression results, the PTE.BW was found positively associated with pain intensity in the MOD pain group, whereas negatively associated with pain intensity in the SEV pain group. The former was given interpretation in the Discussion section. But authors did not give further interpretation on the two incompatible results.
Since, the incompatibility may suggest that the effect of the pain on motor control, and mechanism of pain perception are quite different for the MOD pain and SEV pain groups, which could a very important finding of this study, further explanation is required.
Any information useful for the understanding and understanding of the phenomenon, e.g., how long the subject has suffered from the pain, needs to be provided and explained.
Author Response
REVIEWER #5
It is a paper about authors’ research efforts to make clear the relationship between the chronic low back pain and peak torque, endurance of the trunk flexor and extensor muscles in military police officers (MPO). Experiments were done for collecting data from the MPO groups without pain, with moderate pain and with severe pain. Statistical test was performed to check the difference between the groups. And multivariate regression analysis was performed to identify the relationship between the pain and the target torque and endurance test values.
The data collection and the statistics used are appropriate. The results have been well presented. The topic should be of interest to the readers of the journal.
However, additional information of data collection process and data collected, and/or further analysis are required for interpretation of the results, which are important for the understating of the CLBP of MPO groups. Here are two concerns.
1) The strength and endurance tests might be seriously affected by the pain
This was noted by the authors themselves at the end of the Discussion section, too. Since the subjects were asked to report “adverse pain signs”, it should be possible for the authors to show the reported “adverse pain signs” for different pain groups. Especially for the severe pain group, the report during the tests, shall be very informative for the readers to understand the results.
And since the VAS of the pain and tests have been performed on different days, the pain perception on the day of the tests might be more important. Any information collected on the day of tests might help understand the real nature of the relationship between the pain and peak torque, endurance of the trunk flexor and extensor muscle.
2) Difference between moderate pain and severe pain groups needs further explanation
In the regression results, the PTE.BW was found positively associated with pain intensity in the MOD pain group, whereas negatively associated with pain intensity in the SEV pain group. The former was given interpretation in the Discussion section. But authors did not give further interpretation on the two incompatible results.
Since, the incompatibility may suggest that the effect of the pain on motor control, and mechanism of pain perception are quite different for the MOD pain and SEV pain groups, which could a very important finding of this study, further explanation is required.
Any information useful for the understanding and understanding of the phenomenon, e.g., how long the subject has suffered from the pain, needs to be provided and explained.
ANSWERS
Initially, we want to use this opportunity to thank the reviewer for his time and efforts to provide us relevant feedback. We hope we have resolved all issues in our point-by-point response. The main issues raised by the reviewer were addressed bellow.
We agree with the reviewer that pain is a confounding factor, but, so far, no studies have proposed a methodological approach to overcome such delicate issue. Therefore, we have acknowledged this as a study limitation. Even though we have clearly requested participants to report adverse pain signs, none referred a discomfort that impeded to perform the assessments maximally. They also confirmed that they felt their performances as maximal. To accommodate the reviewer’s concern, we have adjusted and rephrased a sentence bellow in the methods section:
“No discomfort prevented participants from concluding the tests. They were confident that a maximal performance was achieved in all trials.”
The reviewer also raised a relevant question regarding the instant the pain was assessed. We assume that we have misreported the procedures as we also assessed the VAS immediately before the muscle functioning tests. We did not find any differences that required a reallocating the participant to another group. We have also clarified that regression procedures were based on the VAS obtained immediately the peak torque and endurance tests. We apologize for omitting this information in the first version of the manuscript.
“Pain intensity was also assessed immediately before the peak torque and endurance tests. The small variations between the instants pain was assessed were not large enough to demand reallocating a participant to another group.”
“The EVA used as a dependent variable was that measured immediately before the peak torque and endurance tests.”
The reviewer spotted a relevant point regarding the conflicting results between groups. We have no other option unless to propose some speculating arguments that are out of the scope of the study. We agree with the reviewer that some mechanisms play a role and decided to indicate postural adjustments and some mediating mechanisms that modify pain perception. To accommodate this relevant concern, we included the following paragraph.
“On the other hand, it is intriguing the negative association between the peak torque of the extensor muscles and pain intensity in the group with severe CLBP. It may be speculated that pain intensity may cause different responses in CLBP patients. It has been demonstrated that subjects with pain are able to develop alternate postural adjustments, as a strategy performed to cope with the pain. These adjustments are accompanied by muscle recruitment changes in which some muscle groups are selective recruited according to a specific pattern. Lorimer and Hodges demonstrated a delayed recruitment of the transversus abdominis/obliquus internus coupled with an increased recruitment of the obliquus externus during a set of trials in which a painful cutaneous stimulation was applied (Lorimer and Hodges, 2005). It was noticed a gradual return to control values when pain was ceased. Thus, other mechanisms may be involved that allow one to increase the pain perception threshold by activating pain modulation mechanisms (Tobaldini et al. 2019). The chronic exposure to severe pain may have produced central adaptations, which require future investigations that are beyond the scope of the present study.”

Round 2
Reviewer 1 Report
I must start from the comment on the way the Authors responded to reviewer's comments. It is a good practice that comments are repeated, and Authors provide their answers below them and not together in one batch. Now it isn't easy to trace what they respond to.
In the reviewer's opinion, the presentation of the study results should be frank and transparent. And decidedly, the desire to have some statistically significant associations should not govern the way the analysis is conducted and the results are presented. Sometimes one should be able to admit that there are no significant associations between analysed variables.
Said this, the regression analysis on a more numerous study sample is superior than the analysis on a small 30-40 subject samples with seven independent variables. Now it is not clear what is the meaning of these predictors and do they really have some effect or became significant in the results of the analysis on too small and not representative samples.
Furthermore, the Authors were not able to explain why the inclusion of persons reporting no or low pain level in the model introduces a bias to the analysis. If we think about reported pain as a continuous variable, such assumption is not clearly justified.
The division of the study sample depending on the arbitrary decision as for levels of reported pain seems to be counterproductive.
The argument that other studies were published with significant deficiencies in the quality of the statistical analysis is not relevant.
The conditions for the multiple multivariate linear regression are more numerous than those mentioned by Authors. Apart from checking the correlations among independent variables, multicollinearity check should also be based on the assessment of VIP and Tolerance values for all variables included in the model.
Reviewer 2 Report
Recommending publication.
Reviewer 5 Report
Appreciate your careful consideration and revision to the draft.
Since in the first version, section 2.1, the following sentence gave the reviewer a strong impression that the VAS has been performed separately.
In the first visit to the laboratory, participants answered a Visual Analogic Pain Scale (VAS) to quantify the pain intensity
With the following sentence, it is clear.
“Pain intensity was also assessed immediately before the
peak torque and endurance tests. The small variations
between the instants pain was assessed were not large
enough to demand reallocating a participant to another
group.”
Moreover, for the conflicting results in terms of PTE.BM in severe pain group, and moderate pain group, authors' consideration can be understood. Hope the discussion can motivate relevant future work.
“On the other hand, it is intriguing the negative association
between the peak torque of the extensor muscles and pain
intensity in the group with severe CLBP. It may be speculated
that pain intensity may cause different responses in CLBP
patients. It has been demonstrated that subjects with pain are
able to develop alternate postural adjustments, as a strategy
performed to cope with the pain. These adjustments are
accompanied by muscle recruitment changes in which some
muscle groups are selective recruited according to a specific
pattern. Lorimer and Hodges demonstrated a delayed
recruitment of the transversus abdominis/obliquus internus
coupled with an increased recruitment of the obliquus
externus during a set of trials in which a painful cutaneous
stimulation was applied (Lorimer and Hodges, 2005). It was
noticed a gradual return to control values when pain was
ceased. Thus, other mechanisms may be involved that allow
one to increase the pain perception threshold by activating
pain modulation mechanisms (Tobaldini et al. 2019). The
chronic exposure to severe pain may have produced central
adaptations, which require future investigations that are
beyond the scope of the present study.”